# Occurrence and Multidrug Resistance of *Campylobacter* in Chicken Meat from Different Production Systems

**DOI:** 10.3390/foods11131827

**Published:** 2022-06-21

**Authors:** Nânci Santos-Ferreira, Vânia Ferreira, Paula Teixeira

**Affiliations:** CBQF-Centro de Biotecnologia e Química Fina-Laboratório Associado, Escola Superior de Biotecnologia, Universidade Católica Portuguesa, 4169-005 Porto, Portugal; n_ferreira09@hotmail.com

**Keywords:** *Campylobacter*, broilers, campylobacteriosis, backyard, chicken, free-range, antibiotic resistance

## Abstract

*Campylobacter* is the leading bacterial cause of diarrheal disease worldwide and poultry remains the primary vehicle of its transmission to humans. Due to the rapid increase in antibiotic resistance among *Campylobacter* strains, the World Health Organization (WHO) added *Campylobacter* fluoroquinolone resistance to the WHO list of antibiotic-resistant “priority pathogens”. This study aimed to investigate the occurrence and antibiotic resistance of *Campylobacter* spp. in meat samples from chickens reared in different production systems: (a) conventional, (b) free-range and (c) backyard farming. *Campylobacter* spp. was detected in all samples from conventionally reared and free-range broilers and in 72.7% of backyard chicken samples. Levels of contamination were on average 2.7 × 10^3^ colony forming units (CFU)/g, 4.4 × 10^2^ CFU/g and 4.2 × 10^4^ CFU/g in conventionally reared, free-range and backyard chickens, respectively. *Campylobacter jejuni* and *Campylobacter coli* were the only species isolated. Distribution of these species does not seem to be affected by the production system. The overall prevalence of *Campylobacter* isolates exhibiting resistance to at least one antimicrobial was 98.4%. All the *C. coli* isolates showed resistance to ciprofloxacin and to nalidixic acid, and 79.5 and 97.4% to ampicillin and tetracycline, respectively. In total, 96.2% of *C. jejuni* isolates displayed a resistant phenotype to ciprofloxacin and to nalidixic acid, and 92.3% to ampicillin and tetracycline. Of the 130 *Campylobacter* isolates tested, 97.7% were classified as multidrug resistant (MDR).

## 1. Introduction

*Campylobacter* is the major bacterial cause of foodborne illness worldwide and cam-pylobacteriosis has been the most commonly reported zoonosis in humans in the European Union (EU) since 2005 [1,2]. In 2020, the number of confirmed cases of campylobacteriosis was 120,946, representing an EU notification rate of 40.3 per 100,000 inhabitants [1]. Several species of *Campylobacter* are able to cause disease but *Campylobacter jejuni* followed by *Campylobacter coli* are responsible for the majority of infections in humans [1,3]. Most human cases of campylobacteriosis are usually auto-limited, only requiring hydration and electrolyte repletion. Symptoms include diarrhea, abdominal pain, fever, malaise and headaches [3]. Hospitalization and antibiotic therapy may be required for high-risk patients such as the immunocompromised, the elderly or patients with more severe diseases [4]. Macrolides and fluoroquinolones are the primary antibiotics to treat serious *Campylobacter* infections [5]. However, due to the high rates of fluoroquinolone resistance among the genus, macrolides became the first-line antibiotic class for the treatment of campylobacteriosis [5,6]. In addition to the economic costs, deaths and morbidities associated with the infection, the worrying emergence of antibiotic-resistant strains has become a serious public health issue. The intensive use of antimicrobial agents in farm animals has been abolished over the years, as research suggests that the emergence of resistant strains originates from the widespread use of antibiotics in livestock [2]. Even though several sources of *Campylobacter* spp. (e.g., water [7], milk [8], pork [4]) have been linked to human infection, poultry remains the main vehicle of *Campylobacter* spp. transmission to humans [1,9,10]. Commission Regulation (EU) 2017/1495 amending Regulation (EC) No. 2073/2005 [11] establishes the regulatory limit for *Campylobacter* in broiler carcasses (chicken neck skin samples can present up to 1000 colony forming unit (CFU)/g), keeping a process hygiene criteria control in food animal production. However, the control of *Campylobacter* contamination throughout the poultry chain, including breeding, loading, transportation, slaughter, packaging, storage and retailing, is extremely complex and despite all the efforts on the development of new control strategies, the reduction in this pathogen to safe levels is still a quest for both science and industry [12]. The consumers’ bad practices during the handling of raw meat and inadequate cooking are considered one of the top causes for campylobacteriosis occurrence [13].

On the other hand, backyard chickens, that have increased in popularity as an alternative poultry rearing and production system, are generally produced without controlling food safety risks. Comparative studies on the occurrence of *Campylobacter* in conventionally reared and backyard chickens are sparse. Therefore, studies addressing this subject are important to identify the differences in food safety risks presented by alternative systems compared with conventional methods. 

This study aimed to investigate the occurrence and antibiotic resistance of *Campylobacter* in raw chicken derived from different poultry production systems: (a) conventionally reared (*n* = 11), (b) free-range (*n* = 10) and (c) backyard chickens (*n* = 11). *Campylobacter* spp. isolates recovered from chicken samples were typed using pulsed-field gel electrophoresis and tested for antimicrobial susceptibility to nine clinically relevant antibiotics from seven different classes.

## 2. Material and Methods

### 2.1. Sample Collection

Raw backyard chicken samples (*n* = 11) were collected from volunteer citizens that produce poultry for private consumption. Conventionally reared (*n* = 11) and free-range (*n* = 10) whole chicken carcasses were conveniently purchased from local supermarkets or butchers located in the Porto region, Portugal, between March 2018 and June 2019. The commercial establishments where the chickens were purchased included three large supermarket chains (including the two main supermarket chains that dominate retail sales in Portugal) and two local street butchers. The samples purchased at these commercial establishments comprised chicken produced by nine different national producers, including the first three market leaders in the national poultry breeding segment. Samples were transported to the laboratory in ice boxes and then kept refrigerated at 4 °C before analysis and processed up to 24 h after collection. 

### 2.2. Detection and Enumeration of Campylobacter spp.

Skin samples collected from whole chicken carcasses were tested for the presence of *Campylobacter* spp. by colony count technique in parallel with detection method according to the specific International Organization for Standardization standards ISO 10272-2:2017 [14] and ISO 10272-1:2017 [15], respectively. In short, skin was detached from the whole chicken carcasses and cut into small pieces using sterile knifes and cutting boards. Twenty-five grams was randomly collected and weighted into a sterile stomacher bag, diluted in a 1:10 (*w*/*w*) proportion with buffered peptone water (BPW; Biokar Diagnostics, Allonne, France) and homogenized for 2 min (BagMixer S, Interscience, Roubaix, France). Ten-fold dilutions were subsequently prepared in 9 mL of sterile quarter strength Ringer’s solution (R1/4; Biokar Diagnostics) and 1 mL of the first decimal dilution (distributed on the surface of two agar plates) or 0.1 mL aliquots of the further decimal dilutions were spread-plated onto modified charcoal cefoperazone deoxycholate (mCCD; VWR International, Radnor, PA, USA) agar plates and incubated for 44 h at 41.5 °C, in microaerophilic conditions (85% nitrogen (N_2_), 10% carbon dioxide (CO_2_) and 5% oxygen (O_2_)). The detection limit of the enumeration method was 10 colony forming units (CFU)/g. 

To test the presence/absence of *Campylobacter*, a pre-enrichment step was performed by diluting 10 g of chicken skin into 1:10 (*w*/*w*) Bolton broth (VWR International), with 5% (*v*/*v*) defibrinated horse blood (Thermo Fisher Scientific, Waltham, MA, USA). After incubation at 41.5 °C for 44 h in microaerophilic conditions, 10 μL of the enriched suspension was transferred onto selective agar mCCD and CampyFood (CFA; bioMérieux, Marcy-l’Étoile, France) agar plates and incubated in the same conditions. Up to five typical colonies from enumeration and detection plates were sub-cultured on Columbia agar (Merck Millipore, Billerica, MA, USA) supplemented with 5% (*v*/*v*) defibrinated horse blood and incubated under microaerophilic conditions for 24 h for further confirmation, including the microscopy of a freshly prepared bacterial suspension, observation of hemolysis after 24 h incubation, oxidase test and growth under aerobic conditions [14,15].

### 2.3. Confirmation of Campylobacter Species

Identification of *Campylobacter* to the species level was determined using a multiplex PCR that discriminates the different species of thermotolerant *Campylobacter*: *C. jejuni*, *C. coli*, *C. lary*, *C. upsaliensis* and *C. fetus* [16]. Well-isolated *Campylobacter* colonies were grown on Columbia blood agar (bioMérieux) for 24 h at 41.5 °C in microaerophilic atmosphere. Half loopful of culture was transferred into 20 μL of sterile Tris-EDTA solution, heated at 100 °C for 10 min and diluted with sterile water. DNA amplification was carried out in a total of 25 μL PCR reaction containing the primers described by Wang et al. [16].

### 2.4. Subtyping by Pulse-Field Gel Electrophoresis (PFGE)

A total of 130 *Campylobacter* isolates were typed by pulsed-field gel electrophoresis (PFGE) according to the PulseNet protocol [17]. The DNA was digested with SmaI (40 U at 25 °C for 2 h) and KpnI 40 U at 37 °C for 5 h (Thermo Fisher Scientific). *Salmonella* Braenderup H9812 plugs restricted with XbaI were used as the molecular weight size standard. Reference strains used as controls were DSM 4688 (*C. jejuni*) and DSM 4689 (*C. coli*). PFGE was performed using the CHEF DR III System (Bio-Rad, Hercules, CA, USA). Run time was 19 h and 18 h for plugs restricted with SmaI and KpnI enzymes, respectively. The gels were stained with ethidium bromide solution (MP Bio-medicals, Santa Ana, CA, USA) and the DNA banding pattern was captured with the Gel Doc XR+ System with Image Lab Software (Bio-Rad Laboratories, Hercules, CA, USA). BioNumerics v.7.6.2 (Applied Maths, Sint-Martens-Latem, Belgium) was used for numerical analysis of the DNA macrorestriction patterns. Classification of isolates into different SmaI and KpnI patterns was visually validated, and a similarity threshold of ≥98% was used to define isolates belonging to the same PFGE types. 

### 2.5. Antimicrobial Susceptibility Testing 

The susceptibility to nine relevant antimicrobials was carried out by using the disk diffusion method according to the EUCAST (European Committee for Antimicrobial and Susceptibility Testing) guidelines on Mueller–Hinton agar (Biokar Diagnostics), supplemented with 5% (*v*/*v*) defibrinated horse blood (MH-F) and 20 mg/L of β-NAD (Sigma-Aldrich, Darmstadt, Germany). The following antibiotics and concentrations were selected: ampicillin (AMP, 10 μg), amoxicillin/clavulanic acid (AMC, 30 μg), gentamicin (CN, 10 μg), ciprofloxacin (CIP, 5 μg), erythromycin (E, 15 μg), tetracycline (TE, 30 μg), imipenem (IMP, 10 μg), meropenem (MEM, 10 μg) and nalidixic acid (NA, 30 μg). All antimicrobial agents were from Oxoid (Oxoid, Basingstoke, UK). After incubation of the MH-F agar plates at 37 °C for 24 h, under a microaerophilic atmosphere (GENbag microaer, bioMérieux), the growth inhibition halos around each antibiotic disk were read. For ciprofloxacin, erythromycin and tetracycline, the EUCAST breakpoints were used to classify strains as susceptible or resistant (V9.0; January 2019; accessed 19 January 2021, http://www.eucast.org/clinical_breakpoints/). For the other molecules the cut-offs of the Comité de l’Antibiogramme de la Société Française de Microbiologie (CA-SFM) (V.2.0; September 2018) were considered. *Campylobacter jejuni* ATCC 33560 was used as a quality control strain, as recommended by EUCAST. Isolates non-susceptible to three or more antibiotics from different classes were classified as multidrug resistant (MDR) [18].

## 3. Results

### 3.1. Occurrence of Campylobacter spp. among Chicken Samples Derived from Different Production Systems

A total of 32 chicken skin samples from conventionally reared (N = 11), free-range (N = 10) and backyard chickens (N = 11) were analyzed for the presence of *Campylobacter* spp. Results are presented in Table 1 and Table 2. Overall, 29 out of the 32 samples (90.6%) tested positive for *Campylobacter* by at least one of the methods used (i.e., detection and enumeration). All the samples obtained from conventionally reared and free-range broilers were contaminated (Table 1), while in backyard chicken samples, three out of 11 were negative for *Campylobacter* (Table 2). Occasionally it was possible to enumerate *Campylobacter* but not to the detect its presence in the same sample tested. This was probably due to the high number of the contaminating colonies observed on the spread plates of the enrichment broth (detection method) that rendered very difficult to recover isolated *Campylobacter* colonies for further confirmation tests. 

Levels of contamination were on average 2.7 × 10^3^ CFU/g, 4.4 × 10^2^ CFU/g and 4.2 × 10^4^ CFU/g in conventionally reared, free-range and backyard chickens, respectively. One hundred and thirty *Campylobacter* isolates were recovered from detection and enumeration techniques, comprising two species: *C. coli* (*n* = 78) and *C. jejuni* (*n* = 52). Eleven out of the twenty-nine contaminated samples were colonized with both species, while in eleven and eight samples only *C. coli* or *C. jejuni* were recovered, respectively.

### 3.2. PFGE Analysis

The 130 *Campylobacter* spp. isolates recovered from conventionally reared (*n* = 50), free-range (*n* = 46) and backyard (*n* = 34) chicken skin samples were typed by PFGE using two restriction enzymes (SmaI and KpnI). PFGE macrorestriction patterns obtained for all the isolates are given in Appendix A. A high genetic variability was observed amid *C. jejuni* as 35 of the 52 isolates (approximately 73%) presented a unique PFGE type (Appendix A). Fourteen isolates were distributed among four clusters, each of them comprising two to five isolates recovered from the same sample; four isolates were untypable by both enzymes, repeatedly generating incomplete restriction patterns. Conversely, the PFGE analysis of *C. coli* yielded a high number of clusters (Appendix A). For 21 isolates it was not possible to obtain restriction patterns using the KpnI enzyme. Fifty out of seventy-eight isolates (64%) were allocated into seventeen clusters comprising two to four isolates. Fifteen clusters were composed of isolates belonging to the same chicken sample, and only two clusters contained isolates collected from different samples: (i) one cluster comprising two isolates of backyard chicken samples BY2 and BY3 and (ii) one cluster formed by isolates from backyard chicken sample BY5, and free-range chicken samples GC3 and GC4 (originated from different producers and collected in different supermarkets). From the 29 samples positive for *Campylobacter*, 14 (48%) were colonized with more than one strain of *C. jejuni* and/or *C. coli*. 

### 3.3. Antimicrobial Susceptibility Patterns

The 130 *Campylobacter* samples were tested for antimicrobial susceptibility to nine relevant antibiotics. The antimicrobial susceptibility phenotypes observed for *C. jejuni* and *C. coli* isolates are shown in Table 3. The detailed distributions of the antimicrobial susceptibility patterns of *C. jejuni* and *C. coli* isolates among the different production systems, i.e., conventionally reared, free-range and backyard chicken samples, are available in Appendix A.

Overall, there were no differences in terms of antimicrobial susceptibility patterns among isolates from chickens raised under the three different production systems. In addition, isolates from both species exhibited similar phenotypes. A 100% susceptibility was observed to gentamicin, imipenem and meropenem. All the *C. jejuni* isolates and ca. 80% of *C. coli* isolates were also susceptible to amoxicillin/clavulanic acid. The majority of the isolates were also susceptible to erythromycin (90.4 and 73.1% of *C. jejuni* and *C. coli* isolates, respectively). Both species presented a high-level resistance to the remaining four antibiotics tested, i.e., ciprofloxacin, nalidixic acid, tetracycline and ampicillin. All the *C. coli* isolates showed resistance to ciprofloxacin and to nalidixic acid, and 79.5 and 97.4% to ampicillin and tetracycline, respectively. In total, 96.2% of *C. jejuni* isolates displayed a resistant phenotype to ciprofloxacin and to nalidixic acid, and 92.3% to ampicillin and tetracycline.

Only two *C. jejuni* isolates were susceptible to all the antibiotics tested. The remaining 128 isolates were distributed among nine resistance phenotypes (Table 4), including eight profiles of resistance to three or more antibiotics of different classes, being considered MDR bacterial isolates [18]. A MDR phenotype was observed in 99% of the isolates, including all *C. coli* isolates. For *C. jejuni* isolates, resistance to ampicillin, ciprofloxacin, nalidixic acid and tetracycline (AMP^R^-CIP^R^-NA^R^-TE^R^) was the most common profile in samples from the three production systems. This was also the most common profile for most *C. coli* isolates, although a greater variability in resistance profiles has been observed when compared with those displayed by *C. jejuni*.

## 4. Discussion

*Campylobacter* remains the leading foodborne pathogen isolated from poultry. Additionally, the rapid increase in *Campylobacter* antibiotic resistance has led the WHO to place *Campylobacter* fluoroquinolone resistance as a high priority in the WHO Priority Pathogens List to guide research and development of new antibiotics [19]. 

This study illustrates the occurrence and antibiotic resistance of *Campylobacter* in conventionally reared, free-range and backyard chickens. The overall occurrence of *Campylobacter* obtained in this study (90.6%) is comparable to other reports [20,21,22], including to what has been described in Portugal [23,24]. All the samples from chickens raised in conventional and free-range systems were contaminated, while 73% of samples from backyard chickens were positive, suggesting a lower occurrence of *Campylobacter* in the latter. However, the limited number of samples tested does not allow us to draw statistical inferences. A recent meta-analysis study reported no differences between pathogen occurrence among chicken samples from conventional and alternative production systems [25]. Another study also found no differences in *Campylobacter* occurrence between samples from organic and conventional rearing systems [26]. 

Conventionally reared and free-range chickens are subjected to process hygiene criteria during several steps of the production chain. On the other hand, the control of food safety hazards during growth and slaughter of domestic backyard chickens is inexistent. The popularity of this alternative method of producing chicken meat is increasing among consumers. Many believe that the meat of these birds is healthier and safer than that of birds raised by conventional methods due to the reduced use of antibiotics, hormones and pesticides [27,28], even though there is no scientific evidence to support this hypothesis [29,30,31]. A limited number of studies to date have investigated the prevalence of *Campylobacter* in chickens from this type of production system. A high prevalence (86%) has been reported in feces from backyard poultry in New Zealand [32], while a recent study in Australian backyard poultry [33] revealed that only 10% of fecal samples from the flocks were positive.

In our study, the backyard chicken samples presented the lowest occurrence of *Campylobacter*; however, the highest contamination levels were detected in samples from this type of production. Free-range chickens had the lowest levels of *Campylobacter.* This was the only group in which all samples presented a *Campylobacter* contamination level below the process hygiene criteria limit of 1.0 × 10^3^ CFU/g. Conventionally reared broilers and backyard chickens presented levels of contamination above the limit set by the European legislation for process hygiene criteria, which poses a risk for *Campylobacter* foodborne illness. The differences in the level of contamination observed among the three types of production systems may be related to distinct practices during chicken rearing. Free-range chickens are generally from slow-growing breeds, raised outdoors and fed on cereals. In addition, a fewer number of animals are reared together, which means that they have more space when compared with the intensive production systems of the conventional rearing that house a large number of birds and where the control of *Campylobacter* is a challenge, despite the numerous control points and interventions applied [9]. In the case of backyard chickens, the level of contamination of the carcasses can be associated with (a) the cleaning and sanitation procedures applied to the chicken coop and (b) practices during home slaughter, particularly at the time of evisceration of the animal, where a cut or perforation of the stomach and intestine can occur accidentally. Frequently, people that breed animals for private domestic consumption lack the proper knowledge to effectively prevent the contamination of the chicken carcasses by *Campylobacter* or other bacteria. On the other hand, contrarily to what happens in commercial rearing and industrial abattoirs, the number of animals raised together and slaughtered at the same time is usually small, thus the spread of *Campylobacter* contamination is more restrained.

Similar to previous studies, which reported *C. jejuni* and *C. coli* as the most prevalent species in poultry [34,35], these were the only species found in this study. A high genetic variability was observed among 52 *C. jejuni* and 78 *C. coli* isolates recovered, with 45 and 37 unique macrorestriction patterns identified by PFGE, respectively. Some isolates were untypable with one or both enzymes and this phenomenon has been described by other authors [36,37]. It was observed that almost 50% of the chicken samples were contaminated with both species and occasionally with different strains of the same species, which is in accordance with previous reports [38,39,40]. 

Adaptation of *Campylobacter* to the food-producing environments where antibiotics are frequently used has been associated with the development of antibiotic resistance. Quinolone-resistant *Campylobacter* has been described worldwide at alarming rates. As a consequence, macrolides are presently recommended as the first-line therapy of human campylobacteriosis [41,42]. Several studies have reported high rates of quinolone resistance among *Campylobacter* isolated from retail food samples [5,43,44]. In this study, a high rate of resistance was observed for quinolones (98.5% to nalidixic acid and ciprofloxacin) and tetracyclines (95.4% to tetracycline). In total, 20% of the *Campylobacter* isolates were resistant to erythromycin, an antibiotic of the macrolides class. These isolates (including five *C. jejuni* and 21 *C. coli* isolates belonging to different PGE types) were also resistant to ciprofloxacin, which means that the first-choice antibiotics would not be effective. Backyard chickens raised at home are typically not exposed to antibiotics during their development. It would be expected that *Campylobacter* spp. isolated from this type of chicken would be susceptible to most antimicrobials. However, we detected very high antimicrobial resistance rates in *Campylobacter* isolates recovered from samples of the three production systems, highlighting the potential of chicken colonization with *Campylobacter* from the direct environment and surrounding animals [45] as it has been extensively reported in poultry farms [9]. In addition, 99% of *Campylobacter* spp. isolates were resistant to multiple antimicrobial families. High rates of MDR phenotypes have been previously described for *Campylobacter* isolates of human and animal origin in Portugal [46]. The most frequent MDR phenotype (AMP-CIP-NAL-TET) was detected in 53% of the isolates and it has been previously described by Iglesias-Torrens et al. [44] as the most common resistance phenotype in humans and broilers.

## 5. Conclusions

Despite the low number of samples analyzed, results presented in this research clearly indicate that chicken meat from conventional, free-range and backyard farming display high levels of *Campylobacter* contamination. Our findings demonstrate that, contrary to what is generally believed by the consumer, meat from chicken flocks domestically grown presents the same food safety hazards as chicken meat from commercial production systems. The high rate of ciprofloxacin resistance and the observed high level of MDR *Campylobacter* isolated from chicken meat is a serious public health issue requiring interventions at multidisciplinary levels. Therefore, similar to what has been pointed out by other researchers, our study demonstrates that poultry meat is a potential vehicle for the spread of antibiotic-resistant *Campylobacter* in the community. This seems to be a problem shared by the various production systems. Notwithstanding the efforts made by the industry in recent decades, control of contamination in poultry farms remains a serious problem and one that has proved extremely difficult to overcome. Alternative methods to the use of antibiotics are definitely needed, as well as new interventions for *Campylobacter* control in farms and slaughterhouses. However, at this point, the approach that seems to be the most feasible to prevent disease is through the education of the community. It is critical to develop and implement education programs aiming to change consumer behaviors. Poultry meat needs to be perceived as a high-risk food and safe practices need to be adopted by those raising backyard chickens and by the general consumer during the handling and cooking of raw meat.

## Figures and Tables

**Table 1 foods-11-01827-t001:** Occurrence of *Campylobacter* spp. and levels of contamination of samples from conventionally reared and free-range production systems.

Production-System	Shopping Place	Producer	Sample Code	Detection (in 10 g)	Enumeration (CFU/g)	Species Identification
Free-range	Supermarket chain A	P1	C2	Absent	3.0 × 10^2^	*C. jejuni* and *C. coli*
	Supermarket chain A	P2	GC1	Present	6.0 × 10^2^	*C. jejuni*
	Supermarket chain A	P2	GC2	Absent	1.0 × 10^3^	*C. jejuni* and *C. coli*
	Supermarket chain B	P3	GC3	Present	<10	*C. coli*
	Supermarket chain C	P4	GC4	Present	8.10 × 10^2^	*C. coli*
	Supermarket chain C	P5	GC5	Present	2.0 × 10^2^	*C. jejuni* and *C. coli*
	Supermarket chain A	P5	GC6	Present	1.0 × 10^2^	*C. jejuni* and *C. coli*
	Butcher shop D	Unknown	GC7	Present	<10	*C. coli*
	Butcher shop F	Unknown	GC8	Present	1.0 × 10^2^	*C. jejuni* and *C. coli*
	Butcher shop F	Unknown	GC9	Absent	4.0 × 10^2^	*C. jejuni*
Conventionally reared	Supermarket chain A	P4	RT1	Present	1.0 × 10^4^	*C. coli*
	Supermarket chain A	P4	RT2	Absent	1.0 × 10^2^	*C. jejuni*
	Butcher shop E	P6	RT3	Absent	6.0 × 10^3^	*C. coli*
	Butcher shop E	P7	RT4	Present	1.0 × 10^1^	*C. coli*
	Supermarket chain B	P8	RT5	Present	3.7 × 10^3^	*C. coli*
	Butcher shop E	P6	RT6	Absent	6.0 × 10^3^	*C. jejuni*
	Supermarket chain C	P2	RT7	Absent	1.8 × 10^2^	*C. jejuni* and *C. coli*
	Supermarket chain A	P1	RT8	Present	4.2 × 10^2^	*C. jejuni* and *C. coli*
	Supermarket chain C	P9	RT9	Absent	6.3 × 10^2^	*C. jejuni*
	Supermarket chain C	P2	RT10	Absent	2.6 × 10^3^	*C. jejuni*
	Supermarket chain C	P8	RT11	Present	1.6 × 10^2^	*C. coli*

**Table 2 foods-11-01827-t002:** Occurrence of *Campylobacter* spp. and levels of contamination of samples from the backyard production system.

Backyard Farm	Sample Code	Detection (in 10 g)	Enumeration (CFU/g)	Species Identification
1	BY1	Absent	<10	n.a.
2	BY2	Present	3.1 × 10^3^	*C. coli*
3	BY3	Present	<10	*C. coli*
4	BY4	Present	2.5 × 10^5^	*C. jejuni*
5	BY5	Present	2.0 × 10^1^	*C. jejuni* and *C. coli*
6	BY6	Absent	<10	n.a.
7	BY7	Absent	<10	n.a.
8	BY8	Absent	1.0 × 10^1^	*C. jejuni*
9	BY9	Present	8.8 × 10^2^	*C. jejuni* and *C. coli*
10	BY10	Present	7.2 × 10^1^	*C. jejuni* and *C. coli*
11	BY11	Present	<10	*C. coli*

n.a. not available.

**Table 3 foods-11-01827-t003:** Overall distribution of antibiotic susceptibility of *Campylobacter* by species.

Species	Susceptibility ^a^	Antibiotic ^b^ No. of Isolates (%)								
		AMP	AMC	CN	CIP	E	TE	IMP	MEM	NA
*C. coli*	R	62 (79.5%)	1 (1.3%)	0	78 (100%)	21 (26.9%)	76 (97.4%)	0	0	78 (100%)
	I	6 (7.7%)	15 (19.2%)	0	0	0	0	0	0	0
	S	10 (12.8%)	62 (79.5%)	78 (100%)	0	57 (73.1%)	2 (2.6%)	78 (100%)	78 (100%)	0
*C. jejuni*	R	48 (92.3%)	0	0	50 (96.2%)	5 (9.6%)	48 (92.3%)	0	0	50 (96.2%)
	I	1 (1.9%)	0	0	0	0	0	0	0	0
	S	3 (5.8%)	52 (100%)	52 (100%)	2 (3.8%)	47 (90.4%)	4 (7.7%)	52 (100%)	52 (100%)	2 (3.8%)

^a^ S—susceptible; I—intermediate; R—resistant. ^b^ AMP—ampicillin (10 μg); AMC—amoxicillin/clavulanic acid (30 μg); CN—gentamicin (10 μg); CIP—ciprofloxacin (5 μg); E—erythromycin (10 μg); TE—tetracycline (30 μg); IPM—imipenem (10 μg); MEM—meropenem (10 μg); NA—nalidixic acid (30 μg).

**Table 4 foods-11-01827-t004:** Distribution of resistance phenotypes among *C. jejuni* and *C. coli* strains.

Resistance Phenotype	*C. jejuni*					*C. coli*			
	MDR	Conventional	Free-Range	Backyard	Total	Conventional	Free-Range	Backyard	Total
**AMP^R^ CIP^R^ NA^R^ TE^R^**	Yes	19	14	9	**42**	5	6	15	**26**
**AMP^R^ CIP^R^ NA^R^ E^R^ TE^R^**	Yes	4	1		**5**	7	9	2	**18**
**AMP^R^ AMC^I^ CIP^R^ NA^R^ TE^R^**	Yes					6	6		**12**
**CIP^R^ NA^R^ TE^R^**	Yes			1	**1**	3	2	5	**10**
**AMP^I^ CIP^R^ NA^R^ TE^R^**	Yes					4	2		**6**
**AMP^R^ CIP^R^ NA^R^**	Yes			1	**1**		2		**2**
**AMP^R^ AMC^I^ CIP^R^ NA^R^ E^R^ TE^R^**	Yes					2	1		**3**
**CIP^R^ NA^R^**	No			1	**1**				
**AMP^R^ AMC^R^ CIP^R^ NA^R^ TE^R^**	Yes						1		**1**

AMP—ampicillin (10 μg); AMC—amoxicillin/clavulanic acid (30 μg); CIP—ciprofloxacin (5 μg); E—erythromycin (10 μg); TE—tetracycline (30 μg); NA—nalidixic acid (30 μg).

## Data Availability

All data have been included in the manuscript.

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
