# Peer review of "Occurrence and Multidrug Resistance of *Campylobacter* in Chicken Meat from Different Production Systems"

_foods, 2022, doi:10.3390/foods11131827_

Round 1
Reviewer 1 Report
In methodology, please clarify the type of samples at line 68.
Also, line 74: which type of raw chicken>
in line 77: skin samples from which part of carcase ?
Where your conclusion and recommendation of this study ?
Author Response
Thank you very much for taking the time to review our manuscript and for your comments.
R1: In methodology, please clarify the type of samples at line 68.
The following information was added to the manuscript:
"whole chicken carcasses" R1: Also, line 74: which type of raw chicken The following information was added to the manuscript: "Skin samples collected from whole chicken carcasses"R1: in line 77: skin samples from which part of carcase ?
The following information was added to the manuscript: "In short, skin was detached from whole chicken carcasses and cut into small pieces using sterile knifes and cutting boards. Twenty-five grams were randomly collected and ..."
R1: Where your conclusion and recommendation of this study ? A new section (conclusion) was added to the manuscript.
Reviewer 2 Report
Dear Editor, Dear Authors,
Overall, I found a manuscript interesting and revealing alerting concerns on widespread antimicrobial resistance and multidrug resistance among Campylobacter isolates from different types of production systems. In-spite, that some "old school" methodology such as PFGE were applied findings should be presented to the scientific community.
Despite of limited new scientific information, originality and novelty, this manuscript is well-structured and highlights very important emerging issue as multi-drug resistance in Campylobacter in Portugal which is discussed appropriately and needs to be presented/ published.
Some minor comments, remarks or questions still can be addressed to the Authors:
1) Why specifically PFGE as a method was used to describe genetic diversity of isolates, instead of WGS for instance?
2) What was a rationale behind sampling strategy? How sampling sites were chosen? How representative they are? Please add this description under the chapter Sampling- Sample collection!
3) There is a lack of data Statistical analysis! Please, add this to confirm for example any statistical significant/ non-significant differences observed among different production systems on Campylobacter levels of contamination or resistance patterns!
4) Please, improve or strengthen Your conclusions and statements in the Discussion part supported by Your findings/ results!
Overall, results are interesting and can be published, however careful revision and improvement of the manuscript needed!
Sincerely,
Reviewer # ...
Author Response
Answer: Thank you for your valuable comments. We have carefully reviewed the manuscript and taken in consideration all the reviewers’ suggestions.
Some minor comments, remarks or questions still can be addressed to the Authors:
1) Why specifically PFGE as a method was used to describe genetic diversity of isolates, instead of WGS for instance?
Answer: Although other molecular-based or “next generation” typing methods, would provide more information, such as presence of antimicrobial resistance or virulence genes, PFGE is an affordable technique, it’s highly discriminative, and very useful for comparison between bacterial strains. In addition, we have been using PFGE for Campylobacter typing in previous studies and therefore have established all the laboratorial protocols and necessary equipment.
2) What was a rationale behind sampling strategy? How sampling sites were chosen? How representative they are? Please add this description under the chapter Sampling- Sample collection!
Answer: Authors intended to perform a preliminary study to identify the extent of contamination of Campylobacter in backyard chickens and compare with the levels observed in commercial chicken meat. From previous studies by our research team, we were already aware of the high levels of occurrence of Campylobacter in commercial chicken meat available at retail. We collected 11 backyard chickens from breeders of the laboratory team social networks, such as family friends, neighbor’s or work colleagues. During the period of this study, we collected the remaining 21 samples, of conventional and free-range chicken, in retail shops located in the urban area of Porto, specifically in 3 big supermarket chains (including the two main supermarket chains that dominate retail sales in Portugal - supermarket chain A and C), and 2 local street butchers. The meat purchased at these commercial establishments included chicken produced by 9 different national producers, including the first three market leaders in the national poultry breeding segment (P1, P4 and P7). A brief paragraph with this information was added in the current version.
3) There is a lack of data Statistical analysis! Please, add this to confirm for example any statistical significant/non-significant differences observed among different production systems on Campylobacter levels of contamination or resistance patterns!
Answer: For the comparison on the occurrence of Campylobacter in the three production systems, the authors have chosen not to perform a statistical analysis due to the small sample size per system (n=10 or n=11).
Concerning the statistical analysis of the data on the prevalence of antibiotic resistance, the authors are afraid to infer a statistically significant difference between isolates from the different production systems that may not be necessarily biologically significant as, for instance, backyard samples yielded a lower number of isolates (less positive samples) and also a large number of isolates collected from the same samples share the same PFGE type, meaning that are most probably the same strain.
4) Please, improve or strengthen Your conclusions and statements in the Discussion part supported by Your findings/ results!
Answer: The Discussion has been edited and a Conclusion section was added
Reviewer 3 Report
The manuscript reports comprehensive investigation of Campylobacter on chicken carcasses from different production system. The reviewer would like to suggest the following revision:
Line 22 - Replace "Globally" with "Of 130 Campylobacter isolates tested"
Line 62-63 - Please add brief description of each production system i.e. number of birds (range), housing (open/close), slaughtering practice, etc., and proportion of birds produced by each system in the country i.e. conventionally reared XX%, free-range YY%,
Line 65 - add number of antimicrobial classes
Line 67-72 - Add study period (month/year), and if possible, total number of chicken produced during the study period by each system (sampling frame).
Line 68 - replace "randomly" with "conveniently". Randomly select chicken is close to impossible i.e. assign number to all chicken and use random number to select samples. If I understand correctly, the author pick one chicken from a batch of chicken available on the visit day. This is convenient sampling.
Line 225 - replace "prevalence" with "occurrence"
Line 236-237 If conventionally reared and free-range chickens has better hygiene, why do we detect more occurrence of contamination than backyard chicken (which has higher amount).
Line 271-272 There were 26/130 (20%) of Campylobacter isolates with resistance to Erythromycin. Do all of them also resist to Ciprofloxacin as well? If this is the case, the two first line drugs may not work on these isolates. It would also be interesting to see if they share PFGE profile as well.
Author Response
Answer: Thank you for your valuable comments. We have carefully reviewed the manuscript and taken in consideration all the reviewers’ suggestions
Line 22 - Replace "Globally" with "Of 130 Campylobacter isolates tested"
Answer: The word has been replaced according to the reviewer suggestion.
Line 62-63 - Please add brief description of each production system i.e. number of birds (range), housing (open/close), slaughtering practice, etc., and proportion of birds produced by each system in the country i.e. conventionally reared XX%, free-range YY%,
Answer: Authors agree with the reviewer that the information would be interesting to add to the manuscript, although the chicken carcasses were purchased at local supermarkets or butchers, and that information is not made available, not being possible to be included.
Line 65 - add number of antimicrobial classes
Answer: This information was added upon reviewer’ suggestion (L81-82 in the current version).
Line 67-72 - Add study period (month/year), and if possible, total number of chicken produced during the study period by each system (sampling frame).
Answer: The study period was added in the current version). The authors don’t have the information concerning the total number of chicken produced during this period on the different production systems.
Line 68 - replace "randomly" with "conveniently". Randomly select chicken is close to impossible i.e. assign number to all chicken and use random number to select samples. If I understand correctly, the author pick one chicken from a batch of chicken available on the visit day. This is convenient sampling.
Answer: The word “randomly” has been replaced according to the reviewer suggestion to “conveniently”.
Line 225 - replace "prevalence" with "occurrence"
Answer: The word “prevalence” has been replaced as suggested by the Reviewer.
Line 236-237 If conventionally reared and free-range chickens has better hygiene, why do we detect more occurrence of contamination than backyard chicken (which has higher amount).
Answer: A paragraph was added into the Discussion section to examine the differences observed among the three production systems
Line 271-272 There were 26/130 (20%) of Campylobacter isolates with resistance to Erythromycin. Do all of them also resist to Ciprofloxacin as well? If this is the case, the two first line drugs may not work on these isolates. It would also be interesting to see if they share PFGE profile as well.
Answer: Thank you for your comment. Yes, the 26 isolates are also resistant to ciprofloxacin. They include 5 C. jejuni isolates and 21 C. coli isolates. There is not a common PGGE type among C. coli or C. jejuni isolates. A paragraph was added with this information following the sentence referred by the Reviewer.
Reviewer 4 Report
In this manuscript, Santos-Ferreira and colleagues investigated C. jejuni and C. coli contamination and antibiotic resistance in chicken meat. They found that most of the chicken meat was contaminated with C. jejuni and C. coli and most of the pathogens were resistant to at least one antibiotic. The manuscript is well-written, although the small sample number is a concern. The following are some minor concerns.
1. At line 20, what is “ca.97”? Please double check the rest of text for any typos.
2. At lines 248-255, would the different process facility/procedure between the three chickens contribute to the different contamination rate?
Author Response
Thank you very much for taking the time to review our manuscript and for your comments.
At line 20, what is “ca.97”? Please double check the rest of text for any typos.
replaced by 97.4
At lines 248-255, would the different process facility/procedure between the three chickens contribute to the different contamination rate?
More information added:
The differences on the level of contamination observed among the three types of production systems may be related with distinct practices during chicken rearing. Free-range chicken are a slow-growing breed, raised outdoors and fed on cereals. In addition, a lower number of animals is reared together, which means that they have more space when comparing with the intensive production systems of the conventional rearing, that house a large number of birds and where the control of Campylobacter is a challenge, despite the numerous control points and interventions applied [9]. In the case of backyard chickens, the level of contamination of the carcasses can be associated with (a) the cleaning and sanitation procedures applied to the chicken coop, and, (b) practices during home slaughter, particularly at the time of evisceration of the animal, where a cut or perforation of the stomach and intestine can occur accidentally. Frequently, people that breed animals for private domestic consumption lack the proper knowledge to effectively prevent contamination of the chicken carcasses by Campylobacter, or other bacteria. On the other hand, contrarily to what happens in commercial rearing and industrial abattoirs, the number of animals raised together and slaughter at the same time is usually small, thus the spread of Campylobacter contamination is more restrained.Reviewer 5 Report
In the present paper I carefully reviewed, the Authors aimed to assess the occurrence and antibiotic resistance of Campylobacter spp. in meat samples from chickens reared in different production systems.
I would like to congratulate Authors for the good-quality of their article, the literature reported used to write the paper, and for the clear and appropriate structure.
The manuscript is well written, presented and discussed, and understandable to a specialist readership.
In general, the organization and the structure of the article are satisfactory and in agreement with the journal instructions for authors. The subject is adequate with the overall journal scope.
The work shows a conscientious study in which an exhaustive discussion of the literature available has been carried out.
The Introduction section provides sufficient background, and the other sections include results clearly presented and analyzed exhaustively.
However, as specific comments, with the aim to further improve the quality of the paper.
The Introduction section could be further improved by adding a couple of sentences referring to recently published papers.
I suggest to add a separate Conclusion section ; also, the Authors have to check if alle references have been cited in the text.
Author Response
Thank you very much for taking the time to review our manuscript and for your comments.
The Introduction section could be further improved by adding a couple of sentences referring to recently published papers.
A paragraph was added and the following papers were referred:- Hwang, H.; Singer, R. S. J. Food Prot. 2020, 83, 1137–1148. DOI: 10.4315/JFP-19-527.
- Myintzaw, et al. Food Rev. Int. 2022, 1-15. DOI: 10.1080/87559129.2021.1942487
- Myintzaw, et al. J. Food Saf. 2020, 40(4), e12799. DOI: 10.1111/jfs.12799.
I suggest to add a separate Conclusion section ; also, the Authors have to check if alle references have been cited in the text.
Done as suggested
Round 2
Reviewer 2 Report
My comments and remarks were addressed accordingly. Any additional Reviewer views are welcome!